# State formation across cultures and the role of grain, intensive agriculture, taxation and writing

Christopher Opie ●[1] ✉ & Quentin D. Atkinson ●[2]

The invention of agriculture is widely thought to have spurred the emergence of large-scale human societies. It has since been argued that only intensive agriculture can provide enough surplus for emerging states. Others have proposed it was the taxation potential of cereal grains that enabled the formation of states, making writing a critical development for recording those taxes. Here we test these hypotheses by mapping trait data from 868 cultures worldwide onto a language tree representing the relationships between cultures globally. Bayesian phylogenetic analyses indicate that intensive agriculture was as likely the result of state formation as its cause. By contrast, grain cultivation most likely preceded state formation. Grain cultivation also predicted the subsequent emergence of taxation. Writing, although not lost once states were formed, more likely emerged in tax-raising societies, consistent with the proposal that it was adopted to record those taxes. Although consistent with theory, a causal interpretation of the associations we identify is limited by the assumptions of our phylogenetic model, and several of the results are less reliable owing to the small sample size of some of the cross-cultural data we use.

For more than a century, scholars have debated how and why large-scale complex human societies emerged[1]. Early explanations for the formation of the state suggested that the development of agriculture, which enabled sedentism and surplus production, led to the formation of hierarchies with elites controlling the new states[2,3]. However, the long gap between the development of agriculture and the widespread emergence of states has encouraged a refinement of this view. It is now commonly suggested that it was only intensive agriculture that provided enough surplus for the needs of emerging states[4,5]. The timing for the appearance of agricultural intensification—preceding the establishment of hierarchies—is key to supporting this view. A recent study[6] investigating the co-evolution of intensive resource use and socio-political hierarchy in the Pacific found support for a reciprocal relationship, suggesting that social as well as material factors can drive the emergence of political complexity. Another explanation is that, rather than an agricultural surplus, it was the taxation potential of different crops that was crucial to

state formation[7,8]. Scott[8] argues that cereal grains, which require fixed fields, grow above ground, ripen at a predictable time and are readily stored, provide the ideal crop for tax collection. By contrast, roots and tubers are not easily discoverable, have no fixed ripening time, can be left in the ground until needed and do not store well once harvested. Wheat, barley, millet and more recently rice and maize would therefore have provided the key to state formation because of their taxable potential[8]. Scott[8] also points to the fact that all the earliest states that emerged were based on grain: wheat and barley in Mesopotamia, Egypt and the Indus Valley and millet followed by rice in the Yellow River Valley, and, in the New World, maize became a new state crop. To test this view, Mayshar and colleagues[9] used cross-regional data to show that the cultivation of grain is correlated with hierarchy but not with land productivity, suggesting that it was the taxability of grain rather than its potential for producing a surplus that enabled local hierarchies to exploit this feature of grain to their own advantage.

[1]Department of Anthropology and Archaeology, University of Bristol, Bristol, UK. [2]School of Psychology, University of Auckland, Auckland, New Zealand. ✉e-mail: kit.opie@bristol.ac.uk

In support of the taxation view, Scott[8] also reiterates an earlier argument[10,11] that it was only with the adoption of writing as the means to record taxes due and those paid that an emerging state would have been at all viable. The role of record-keeping in state formation and maintenance has previously been tested by Basu and colleagues[12] using a worldwide data sample, which showed a positive and nonlinear association between community size and record keeping. More recently, Stasavage[13] has used the same dataset to suggest that the availability of writing was positively correlated with the emergence of states.

The quantitative studies[9,12,13] testing the link between crop type and appropriability, writing and state formation have all made extensive use of data from the Standard Cross-Cultural Sample (SCCS)[14]. The SCCS is a subset of 186 cultures from the 1,291 cultures in the Ethnographic Atlas (EA)[15], which includes 1,781 separate variables. The SCCS was originally developed to try to deal with the problem of the non-independence of data through the shared histories of cultures in cross-cultural studies. It was hoped that the problems of proximity and common descent of cultures could be addressed by retaining a single culture for each region. However, the SCCS still shows high levels of both spatial and cultural autocorrelation, which seriously limits the interpretation of statistical analyses of global cross-cultural studies using these data[16]. To identify independent instances of correlated change in traits, a cultural phylogeny is required[17]. This has the added advantage of not only being able to test reliably for correlated evolution between traits across the phylogeny but also enabling the inference of relative timing for the appearance of those traits. If the best-fitting model is one in which the gain or loss of one trait is contingent on the presence or absence of the other, the direction of causation can be inferred[17]. This approach has been successfully applied in a number of studies to model the co-evolution of cultural traits along the branches of language family phylogenies[18–21]. The SCCS is a particularly rich data source for cultures worldwide, which means that the recent development of global language supertrees[22,23] has now unlocked those data for phylogenetic cross-cultural analyses. A phylogenetic approach using a global supertree and drawing on SCCS data has already been used to explore marriage patterns[24], food sharing[25] and genital mutilation[26] worldwide.

Here, we aim to leverage the potential of phylogenetic analysis to test for a correlated evolution between pairs of traits linked to the origin of the state and also to investigate the relative timing of the evolution of those traits to infer the likely direction of causality[17]. We do this using a new posterior treeset of world languages[22], representing the genealogical relationships between cultures, derived using Bayesian phylogenetic inference techniques, matched to 868 cultures in the EA and to the 186 cultures in the SCCS.

One approach is to consider cultural complexity in the form of a set of classificatory criteria[27], spanning technological specialization and social stratification, population density and the use of money (for example, refs. 6,12,28). Another approach is to focus on a single classification, such as political complexity (for example, ref. 29), defined as the number of distinct jurisdictional levels beyond the local community[27] from autonomous bands to petty and larger chiefdoms to states. Here, we take this second approach, testing hypotheses concerning the evolution of political complexity and particularly the move from autonomous bands and chiefdoms to fully functioning states. We use the EA/SCCS variable Jurisdictional Hierarchy Beyond Local Community (EA033, SCCS237) as the measure of political complexity (following refs. 9,13,29), where a non-state is defined as having zero to two jurisdictional levels and a state is defined as having at least three levels. We test a number of hypotheses for the emergence of states worldwide using a Bayesian phylogenetic approach[30]:

(1) The intensification of agriculture, which produces a surplus, is the crucial factor enabling the emergence of states[4,5].
(2) Cereal grain agriculture and its potential for taxation is key to state formation[8,9,13].

(3) Writing is adopted to manage taxation and is key to the emergence and maintenance of the state[8,10–13].

## Results

We matched 868 societies from the EA to the posterior treeset of the Global Supertree developed by Bouckaert and colleagues[22] (Fig. 1). Using EA data on jurisdictional hierarchy beyond the local community (EA033) (Fig. 2), we tested several hypotheses for the emergence and maintenance of states. In tests that included measures of agriculture and jurisdictional hierarchy only, we were able to use the full EA data to maximize the reliability of our estimates. However, data on taxation and writing were only available in the SCCS dataset (a subset of EA data) (Supplementary Fig. 1), with taxation data only available for 83 societies. All traits showed a phylogenetic signal (PhyloD)[31] significantly stronger than expected under a random distribution of data on the tree (Methods; Supplementary Table 2), violating the assumption of non-independence and supporting the use of phylogenetic methods to account for the shared histories of societies. At the same time, all traits, except writing, showed a significant difference from Brownian motion, suggesting that other unmodeled factors may play a role.

### Correlated evolution

To test for correlated evolution between pairs of binary traits under an assumption of phylogenetic inheritance, we used the stepping-stone sampler method[32] within the BayesTraits[30] analyses to compare the log marginal likelihood of the dependent (where traits evolve together) and independent models (where traits are restricted to evolve separately) for each pair of traits (Supplementary Table 3). log Bayes factor values (BF) were calculated for each pair of traits with <2 showing weak evidence, >2 positive evidence, 5–10 strong evidence and >10 very strong evidence[33]. Grain and taxation show positive evidence of correlated evolution (BF of 3.38), non-grain agriculture and the state show strong evidence (BF of 7.51), and the rest of the trait pairs show very strong evidence of correlated evolution.

### Intensive agriculture

Intensive agriculture was present in 241 (28%) societies we sampled. In total, 66 states practiced intensive agriculture (72%), whereas 26 states did not (28%). A total of 163 non-states had intensive agriculture (22%), and 587 did not (78%). Accounting for non-independence due to shared ancestry on the global language phylogeny, we find very strong evidence for correlated evolution between intensive agriculture and the emergence of states (BF of 53.56) (Fig. 4 and Supplementary Table 3). The rate matrix (Fig. 4) shows the eight transitions between the two states of the two traits. Each transition rate (for example, from trait state 0,0 to trait state 0,1, $q_{12}$) has the mean rate across the model iterations and the percentage of iterations with a zero rate (for example, where no transition took place in half of the iterations, $Z = 50\%$). The rate matrix in Fig. 4 indicates that the presence of intensive agriculture makes the emergence of states somewhat more likely (rate of transition to statehood in the presence of intensive agriculture ($q_{34} = 0.06$) is twice the rate of transition to statehood in the absence of intensive agriculture ($q_{12} = 0.03$)). However, we find stronger evidence that the presence of a state makes the transition to intensive agriculture more likely ($q_{24}$ (0.31) is six times the rate of $q_{13}$ (0.05)). We also find strong evidence that the presence of statehood makes the loss of intensive agriculture less likely ($q_{31}$ (0.27) is seven times the rate of $q_{42}$ (0.04)) but no evidence that intensive agriculture sustains statehood ($q_{21}$ (0.31) is equal to $q_{43}$ (0.31)). This result provides limited support for hypothesis 1 that the surplus provided by agricultural intensification was important for the emergence of states. We find stronger evidence that the presence of the state encouraged intensive agriculture and that once a state gained intensive agriculture, it was much less likely to lose it than a non-state.

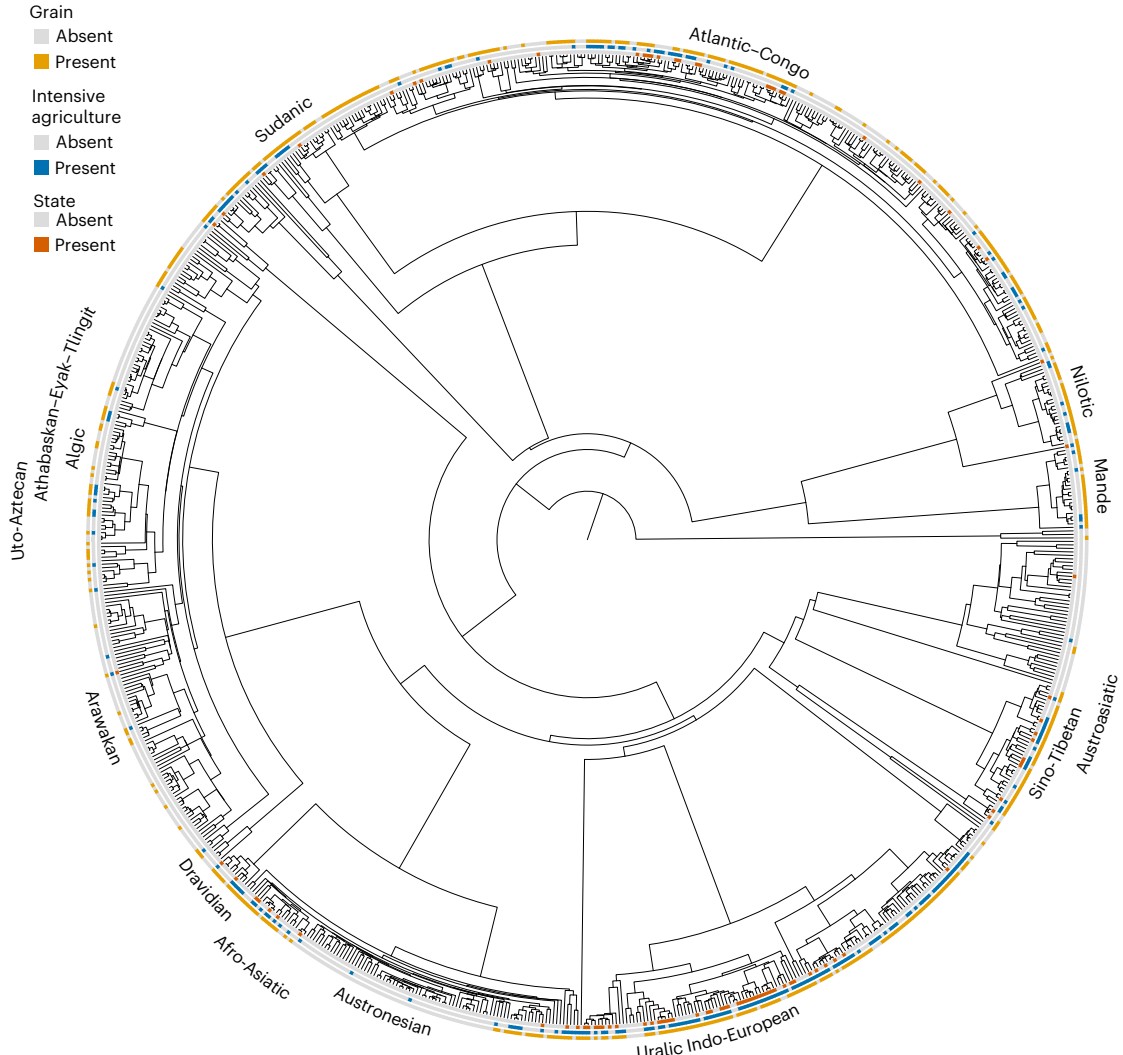

**Fig. 1 | Phylogenetic distribution of data for state, intensive agriculture and grain.** A maximum clade credibility tree of the global treeset[22] with larger language families identified, matched to the data for state, intensive agriculture and grain from the EA. The maximum clade credibility tree is just one tree from the posterior distribution of 1,000 trees. All the analyses we report were performed on the full posterior distribution of trees, integrating over the considerable uncertainty in ancestral relationships between the world's languages. The figure was produced using Treeviewer[44].

## Grain

Grain was the main crop in 484 societies (56%), but not the main crop in 379 societies (44%) (Fig. 3). A total of 76 states depended on grain (84%), whereas 14 states were dependent on other crops (16%). In total, 386 non-states depended on grain (52%), whereas 362 non-states did not (48%). The cultivation of grains shows very strong evidence of correlated evolution with the emergence of states (BF of 22.59) (Supplementary Table 3). The rate matrix in Fig. 5 indicates that the presence of grain agriculture makes the emergence of states possible (the rate of transition to statehood in the presence of grains ($q_{34}$) is low but positive (0.07), whereas the rate of transition to statehood in the absence of grains ($q_{12}$) is zero). In addition, we find evidence that the presence of a state may make the transition to grain agriculture slightly more likely ($q_{24}$ (0.12) is greater than $q13$ (0.07)). We find no evidence that the presence of statehood makes the loss of grain cultivation less likely ($q_{42}$ (0.07) is equal to $q_{31}$ (0.07)) and no evidence that grain agriculture sustains statehood ($q_{21}$ (0.39) is equal to $q_{43}$ (0.39)). This result supports hypothesis 2, which posits that grain agriculture was important for the emergence of states.

Scott[8] argued that there were only grain states, such that no states emerged in societies based on other forms of agriculture. Although our results generally support this pattern, there were a minority of cases in the EA dataset that did not. In 14 of the societies classified as states (16%), grain was not the major crop: 10 had roots or tubers as the main crop, 3 had tree fruits and 1 had vegetables. The majority of these 14 societies were small states[10], and most[9] were in the Atlantic–Congo language family, all based in tropical Africa.

To investigate this further, we pruned the global phylogeny[22] to include only the 241 societies from the Atlantic–Congo language family. We used this new posterior sample of trees to test for correlated evolution between intensive agriculture and the emergence of states, between grain agriculture and the emergence of states and between non-grain agriculture and the emergence of states. In all three cases, there was no evidence of correlated evolution (Supplementary Table 4 and Supplementary Fig. 6), consistent with a scenario in which these agricultural traits did not influence the emergence of states in societies in the Atlantic–Congo language family.

## Non-grain

To check whether other types of agriculture show a similar association with the emergence of states, we tested for correlated evolution between non-grain agriculture (vegetables, tree fruits, roots and tubers) and jurisdictional hierarchy. There was evidence of strong correlated evolution between non-grain agriculture and the emergence of states

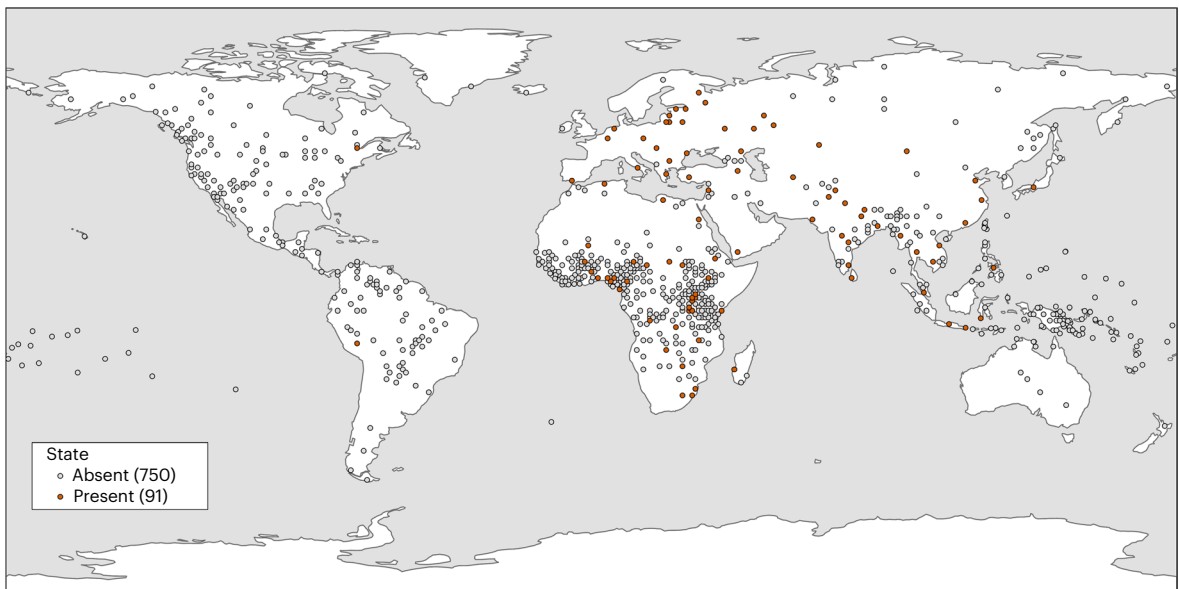

**Fig. 2 | Geographic distribution of states.** A plot of EA data on state and non-state societies. The base map for this figure was obtained from ESRI's World Terrain Reference (https://goto.arcgisonline.com/maps/Reference/World_Reference_Overlay) and generated using ArcGIS Pro 3.5.3[45].

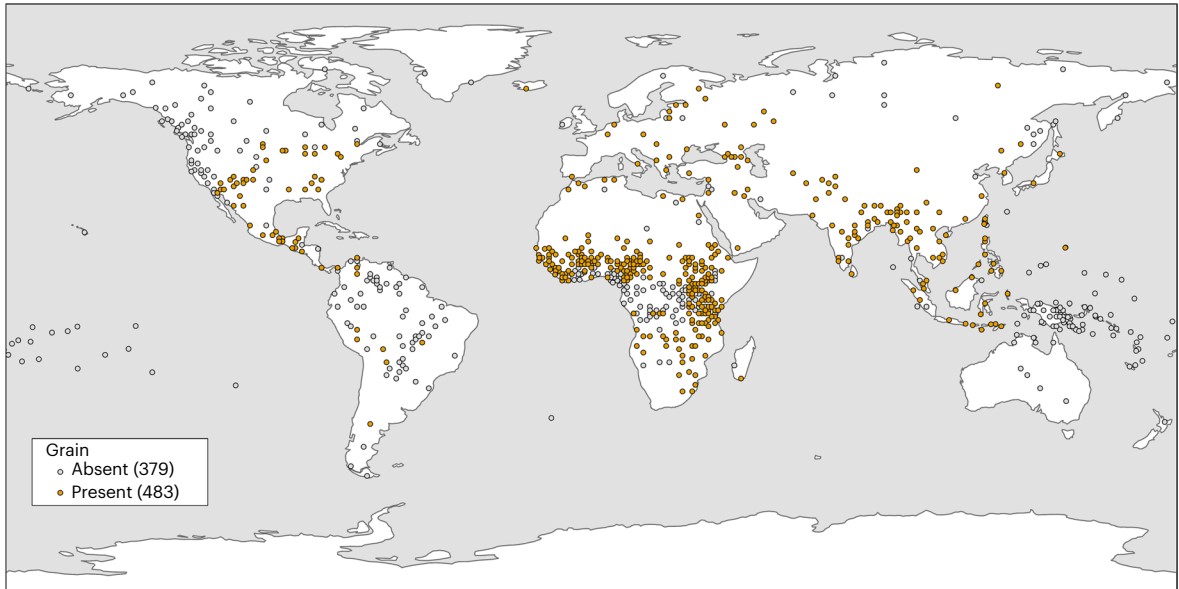

**Fig. 3 | Geographic distribution of grain agriculture.** A plot of EA data on societies with grain present and absent. The base map for this figure was obtained from ESRI's World Terrain Reference (https://goto.arcgisonline.com/maps/Reference/World_Reference_Overlay) and generated using ArcGIS Pro 3.5.3[45].

(BF of 7.51) (Supplementary Table 3 and Supplementary Fig. 7). However, there was no evidence for the emergence of states being more likely in the presence of non-grain agriculture ($q_{12}$ (0.04) equal to $q_{34}$ (0.04)) or that non-grain agriculture was more likely to be gained in states rather than non-states ($q_{13}$ (0.04) equal to $q_{24}$ (0.04)). By contrast, non-grain agriculture was much more likely to be lost in states than in non-states ($q_{42}$ (0.37) is nine times the rate of $q_{31}$ (0.04)). This result suggests that states may have encouraged the growing of grain, while discouraging other forms of agriculture (Supplementary Fig. 7): supporting hypothesis 2 that grain was ideal for taxation.

## Taxation and writing

Next, we tested whether there was correlated evolution between grain agriculture and taxation. Across the SCCS societies, tax was levied in 30 societies (36%) and not levied in 53 societies (64%). In societies where grain was the main crop, tax was levied in 19 (50%) and not levied in 19 (50%). In societies that were not dependent on grain, 10 societies levied taxes (23%), whereas 34 societies did not (77%). The cultivation of grain and taxation show positive evidence for correlated evolution (BF of 3.38) (Supplementary Table 3 and Supplementary Fig. 8). The rate of transition to taxation in the absence of grain agriculture ($q_{12}$) is zero, whereas the rate of transition to taxation in the presence of grain agriculture ($q_{34}$) is high (0.09), suggesting that grain agriculture consistently predicts taxation (Supplementary Fig. 8). There is no evidence that the presence of taxation makes the gain or loss of grain agriculture more or less likely ($q_{24}$ (0.09) and $q_{13}$ (0.09) equal, $q_{42}$ (0.09) and $q_{31}$ (0.09) equal). This result provides further support for hypothesis 2, showing evidence that taxation is more likely to appear in those societies that rely on cereal grain as their main crop. However, the low BF for correlated evolution and the low coverage of data on taxation in the SCCS (83 societies) means that this result may be less robust than previous results.

BF of 53.56

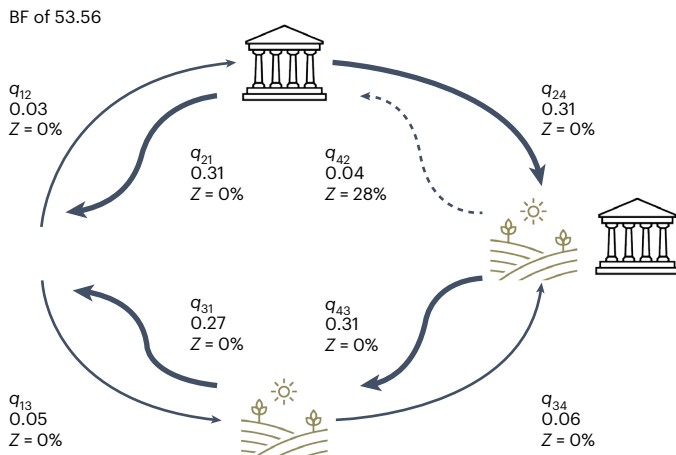

**Fig. 4 | Correlated evolution between intensive agriculture and the emergence of states.** There is very strong correlated evolution (BF of 53.56) between these variables. The width of each arrow is equivalent to the rate of change between the state of the traits.

Writing is present in 38 societies (22%) across the SCCS and absent in 138 (78%). In societies that levied tax, 12 had writing (40%) and 18 did not (60%). In societies without tax, 5 had writing (9%), whereas 48 did not (91%). Our results show very strong correlated evolution between the raising of taxes and the adoption of writing (BF of 19.90) (Supplementary Table 3). The rate of transition to writing in the absence of taxation is zero ($q_{12}$), whereas the rate of transition to writing in the presence of taxation ($q_{34}$) is very high (0.06), suggesting that taxation consistently predicts the adoption of writing (Supplementary Fig. 9). This result supports hypothesis 3, suggesting that writing is adopted in societies that raise taxes, most likely to record those taxes. However, because of the relatively low coverage of data on taxation, we are less certain of this result.

From SCCS data, states are present in 27 societies (16%) and absent in 147 (84%). Writing was present in 21 states (57%) and 16 non-states (43%). Writing was absent in 6 states (4%) and 131 non-states (96%). Support for a model of correlated evolution between the adoption of writing and the emergence of states is very strong (BF of 48.40) (Supplementary Table 3). The rate of transition to statehood in the absence of writing ($q_{12}$) is low but non-zero (0.02), whereas the rate of transition to statehood in the presence of writing ($q_{34}$) is one of the highest rates we observe (0.17). The rate of transition for the adoption of writing ($q_{24}$) is high in states (0.15), whereas in non-states ($q_{13}$) it is low (0.01). Furthermore, a zero rate for the loss of writing in the presence of states ($q_{42}$) suggests that states keep writing once it is adopted (Supplementary Fig. 10). This result supports hypothesis 3, suggesting that writing and state emergence are highly correlated. We find evidence that the presence of writing encourages the emergence of states and states encourage the emergence of writing. Furthermore, once states are established, they are unlikely to lose writing, suggesting that it is important for state maintenance.

### Additional robustness checks

The above findings are based on an assumption of binary trait evolution along the branches of a language phylogeny. One concern with this approach is that more recent unmodelled borrowing between lineages could bias our rate estimates by causing the origins of several traits to be reconstructed too deep in the global tree. For example, grain, statehood and writing are widespread across many of the Indo-European ethnolinguistic groups in our sample (Fig. 1 and Supplementary Fig. 1), but this probably reflects a mix of vertical inheritance and more recent borrowing, rather than simple inheritance of these traits from a Proto-Indo-European ancestor. Defenders of phylogenetic comparative

methods note that such borrowing events can themselves provide insight into the sequence of trait evolution within societies[17] and that the methods are robust to realistic levels of borrowing, consistently outperforming conventional regression approaches[34]. Nevertheless, the reliability of inferences in any given case is likely to depend on a combination of factors, including the rate at which traits are borrowed, their degree of coupling, rates of extinction and sample size.

To incorporate additional historical knowledge into our analyses and to evaluate the robustness of our findings to model misspecification due to more recent borrowing, we repeated each analysis constraining the value of traits at the root of the ten largest language families. When all traits were constrained to be absent at the root of each family, consistent with the assumption that they emerged more recently within each family, we found similar support for correlated evolution to our initial analyses (Supplementary Table 5), with only the correlation for taxation and writing reducing from very strong evidence (BF of 19.90) to strong evidence (BF of 9.15). There was no change in the pattern of transition rate results reported (Supplementary Table 6). Of all the variables we considered, only grain may have already emerged at the root of several of the major global language families[35]. We therefore also reran the two analyses that included grain as a variable, allowing grain to be absent or present at each language family root, while the other binary variable was constrained to be absent. In the grain/state analysis, there was again no change to the results reported with correlation (BF of 23.73) very similar to the unconstrained model (BF of 22.59). In the grain/tax analysis, the support for correlation between the traits showed only weak evidence (BF of 1.41), lower than the positive evidence in the unconstrained model (BF of 3.38) and the model constrained to both traits absent (BF of 2.79). In both cases, the pattern of transition rates remained the same (Supplementary Table 6).

Overall, these analyses give additional support to our main results, suggesting that, although several of the traits we investigate may reflect more recent borrowing, our findings are relatively robust to uncertainty deeper in the phylogeny.

## Discussion

In this study, we have tested a number of hypotheses for the emergence of states across human history. This is part of a wider debate about the movement from small-scale to large-scale complex societies, but here, we focus on the move to centralized bureaucratic states that can have widely differing population sizes[4]. The hypotheses for the emergence of the state include: the intensification of agriculture, which provides

BF of 22.59

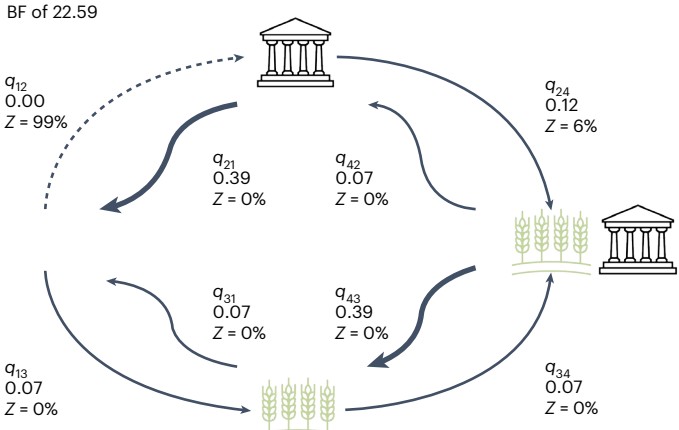

**Fig. 5 | Correlated evolution between grain agriculture and the emergence of states.** There is a very strong correlated evolution (BF of 22.59), with a positive $q_{34}$ rate, but a zero $q_{12}$ rate, suggesting the shift to grain agriculture consistently predicts the emergence of states. The width of each arrow is equivalent to the rate of change between the state of the traits.

a surplus[4,5]; the taxation potential of cereal grain[8,9,13]; and the adoption of writing to record tax and enable and maintain states[8,10–13].

This study uses phylogenetic methods to test these hypotheses for state formation on a global scale. The recent development of global language phylogenies[22,23] makes it possible to use large cultural datasets such as the EA[15] and the SCCS[14] while accounting for the shared histories of cultures in cross-cultural studies[16]. Supporting the need for such an approach and the use of language phylogenies as a plausible model of trait inheritance, all of the variables in our study show evidence of phylogenetic signal. However, variation among the traits in our study is not perfectly captured by the language phylogeny—the Purvis D scores differed significantly from that expected under a strict Brownian diffusion model of trait evolution along the branches of the tree. We therefore acknowledge that other, as yet unmodelled processes, such as environmental factors, are also at work and stress that our findings should be interpreted as contributing to wider literature on the origins of statehood rather than offering the last word.

The first hypothesis tested argues that it was the intensification of agriculture, specifically the use of fertilization or crop rotation to reduce fallow periods and irrigation, which enabled a surplus to be produced that was used to form states[4,5]. Our results indicate that the use of intensive agriculture was indeed tightly coupled with the emergence of states worldwide. However, our findings suggest that although the presence of intensive agriculture makes the emergence of states slightly more likely, it is the inverse causal direction that is the stronger relationship, with the presence of states making the use of intensive agriculture much more likely. This result supports a previous study[6] using phylogenetic methods to show that political complexity was more likely to have driven intensive agriculture than to be a result of it across Austronesian societies. We also found that, among societies that had adopted intensive agriculture, states were much less likely to lose the practice than non-states, suggesting that states may nevertheless have played an important role in the trend towards increasing agricultural intensification.

The emergence of states due to the taxable potential of cereal grains was the second hypothesis we tested. It is argued that grain is particularly good for taxation due to the use of fixed fields and because grain grows above ground, ripens at predictable times and can be stored for very long periods[8]. Our analyses indicate strong support for correlated evolution between the adoption of cereal grains as the main crop by societies and the emergence of states. Furthermore, our analyses suggest that states were very unlikely to emerge in societies without grain production, whereas states were very likely to emerge in societies with cereal grains as their main crop. This result supports those of a previous study[9] using similar data, suggesting that it was the appropriability of grain that led to state formation. The advantage of the phylogenetic methods we deployed here is that we were able to both account for the non-independence of societies due to shared language ancestry and also suggest the direction of causation, contingent on evolution along the branches of a posterior distribution of plausible global language relationships[17]. For comparison, we also tested the hypothesis that it was non-grains, such as vegetables, tree fruit, roots and tubers, that resulted in the formation of states worldwide. The results again indicated strong correlated evolution between non-grain agriculture and state formation but suggest that once formed, states were much more likely to lose non-grain crops than non-states.

In our dataset, there were a few states without grain, of which most had roots or tubers as the main crop. Most of these small states were in tropical Africa. This finding emphasizes the potential for regional variation in state formation, perhaps linked to the role of unmodelled environmental contingencies. For example, environmental factors may have reduced the payoffs of grain for states in Sub-Saharan Africa. Indeed, when restricting our analysis to only the Atlantic–Congo

language family of Sub-Saharan Africa, we found no evidence of correlated evolution between grain and the emergence of states, consistent with the suggestion that environmental factors play an important role[9]. Future work may be able to formally model these effects.

Nevertheless, our findings support a connection between the production of cereal grains and the emergence of states outside Sub-Saharan Africa. The proposed mechanism for this is that grain is ideal for taxation purposes[8,9,13]. Our results also support this argument, although our analysis of taxation data is less robust because it is based on a much smaller dataset. Grain production and taxation show positively correlated evolution worldwide. Furthermore, our findings suggest taxation was less likely to arise in societies without grain production and more likely in those with grain production.

The third hypothesis we tested was the role of writing in the emergence of states via its importance for recording taxation. Taxation requires a trustworthy method of recording taxes, and once states have emerged, writing has been argued to be essential for their maintenance[8,10–13]. Our results indicate that the adoption of writing is, indeed, strongly correlated with both taxation and the emergence of states. We found writing was very unlikely to be adopted in societies that do not raise taxes but very likely in societies that do. States did emerge in societies without writing but were much more likely to emerge in societies with writing. Furthermore, once states have emerged, we found they were very unlikely to lose writing. These results support previous studies[12,13] using the same dataset. However, we are able to show that the relationship holds when accounting for the common linguistic ancestry of societies and that the hypothesized relative timing of trait change is also supported under our model. Again, these results are based on the smaller sample size of the SCCS, with taxation being a particularly small dataset.

Drawing strong inferences about human prehistory is an inherently challenging task, particularly when it comes to establishing causal relationships between complex phenomena such as modes of agricultural production and statehood. With this in mind, we want to emphasize that the findings we have presented here are contingent on the reliability of the available cross-cultural data, and on the assumptions of our model, most notably the extent to which binary trait evolution along the branches of a language phylogeny is an accurate description of the processes at work. The model we deploy is, like any model, a simplification and does not, for example, explicitly incorporate horizontal transmission or the role of environmental factors in constraining or canalizing social evolution. Nevertheless, there are good reasons to hold our findings credible. The consistent phylogenetic signal in the data we consider suggests a phylogenetic model is a reasonable approximation and an important extension on prior work that has not sought to model historical dependencies between societies. To the extent that our data reflect a more recent borrowing of traits, such events are themselves a legitimate source of change down cultural lineages[17] and the methods we deploy have been shown to be robust to realistic levels of borrowing, outperforming the conventional regression techniques used in prior work[34]. Moreover, our findings incorporate considerable phylogenetic uncertainty across a posterior distribution of global language trees, suggesting that our inferences are not sensitive to a specific language tree topology, particularly for deeper relationships in the tree, where phylogenetic uncertainty is greatest. In addition, when we constrain ancestral states of major language families to incorporate historical knowledge consistent with a more recent spread of traits, our inferences are also unaffected.

Together, then, the overall pattern of results we present shows a clear concordance with and support for a newly emerging picture of the origin of states across the world. Namely, that it was not the surplus production of agricultural intensification but the taxable nature of cereal grains that led to both the emergence of states and the adoption of writing.

## Methods

The variables used were as follows:

(1) Jurisdictional hierarchy beyond the local community (EA033 and SCCS237): (1) autonomous bands (no levels), (2) petty chiefdoms (one level), (3) larger chiefdoms (two levels), (4) small states (three levels) and (5) large states (four levels). Here, states include both small and large states, with at least three jurisdictional levels above the local community.

(2) Agriculture: intensity (EA028), with intensive agriculture defined as using fertilization, crop rotation or other techniques to shorten or eliminate the fallow period and irrigation.

(3) Agriculture: major crop type (EA029 and SCCS233), cereal grain. Non-grain was defined as vegetables, tree fruit, roots and tubers.

(4) Taxation paid to community (SCCS784), regular taxes[36].

(5) Writing and records (SCCS149), true writing, with or without records[27].

The variables were made binary to test the three hypotheses directly (see Supplementary Table 1 for further detail on variables used and the criteria used for binary coding).

### Phylogeny

The relationships between the world's languages, particularly deeper macro-relationships between established families, and the timing of diversification events, are highly contentious. A common response to this has been to avoid global phylogenetic analyses altogether or to use crude proxies for ancestry such as language family membership to attempt to control for non-independence and model change. Here, we take a different approach, using a newly available Bayesian posterior treeset representing a global super tree of the worlds' languages developed by Bouckaert and colleagues[22]. The treeset has been used in a number of studies[37–39], and the posterior distribution and code used to generate it are available[40]. The supertree was pruned to the cultures included in the EA and the SCCS using Phytools in R[41]. This posterior treeset of 1,000 trees is derived from prior information on what is known about the sequence and timing of the breakup of the world's languages, including specifying the considerable uncertainty in branch lengths and tree topology. This provides a principled statistical framework for modelling cultural evolution on the global tree, while integrating out phylogenetic uncertainty from our inferences.

### Analyses

The correlated evolution analyses were conducted using the Discrete model in BayesTraits V3.0.5[42], using a reverse-jump hyperprior approach with an exponential hyperprior (0–1.0), allowing the priors to be estimated from the data[30]. Models were run for 110,000,000 iterations with the first 10,000,000 iterations discarded as burn-in. This approach estimates support for correlated evolution using a stepping-stone sampler method[32] to calculate the log BF for the likelihood of the dependent model over the independent model. Log BF values were calculated such that <2 shows weak evidence, >2 positive evidence, 5–10 strong evidence and >10 very strong evidence[33].

### Transition rate matrices

The model results also show the level of support for the transition rates between the two states of the two variables in terms of the mean rate and the percentage of model iterations that show a transition rate of zero[30]. The rate matrix figures show the eight transitions between the two states of the two traits. Each transition rate (for example, $q_{12}$) has the mean rate across the model iterations and the percentage of iterations with a zero rate (for example, $Z = 50\%$).

### Reporting summary

Further information on research design is available in the Nature Portfolio Reporting Summary linked to this article.

## Data availability

The data used in these analyses are taken from the EA[15] and the SCCS[14], a subset of 186 cultures from the EA, downloaded (September 2022) from the public database D-Place[43]. The maps throughout this paper were generated using ArcGIS Pro 3.5.3 and ESRI's World Terrain Reference.

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

## Acknowledgements

C.O. received funding for an Early Career Fellowship (ECF 619) from the Leverhulme Trust for part of the research. Q.D.A. was partly supported by two Royal Society of New Zealand Marsden grants (grant nos. MFP-20-UOA-123 and MFP-24-UOA-126) and a grant from the Templeton Religion Trust (grant no. TRT-2022-30666). We thank T. Currie and S. Montgomery for useful advice on analysis and interpretation and M. Gillings for help producing the maps.

## Author contributions

C.O. and Q.D.A. designed and performed the research. C.O. collected data. C.O. analysed data. C.O. and Q.D.A. wrote the paper.

## Competing interests

The authors declare no competing interests.

## Additional information

**Correspondence and requests for materials** should be addressed to Christopher Opie.

# Reporting Summary

## Statistics

For all statistical analyses, confirm that the following items are present in the figure legend, table legend, main text, or Methods section.

| n/a | Confirmed | |
|---|---|---|
| ☐ | ☒ | The exact sample size (*n*) for each experimental group/condition, given as a discrete number and unit of measurement |
| ☐ | ☒ | A statement on whether measurements were taken from distinct samples or whether the same sample was measured repeatedly |
| ☐ | ☒ | The statistical test(s) used AND whether they are one- or two-sided *Only common tests should be described solely by name; describe more complex techniques in the Methods section.* |
| ☐ | ☒ | A description of all covariates tested |
| ☐ | ☒ | A description of any assumptions or corrections, such as tests of normality and adjustment for multiple comparisons |
| ☒ | ☐ | A full description of the statistical parameters including central tendency (e.g. means) or other basic estimates (e.g. regression coefficient) AND variation (e.g. standard deviation) or associated estimates of uncertainty (e.g. confidence intervals) |
| ☒ | ☐ | For null hypothesis testing, the test statistic (e.g. *F*, *t*, *r*) with confidence intervals, effect sizes, degrees of freedom and *P* value noted *Give P values as exact values whenever suitable.* |
| ☐ | ☒ | For Bayesian analysis, information on the choice of priors and Markov chain Monte Carlo settings |
| ☒ | ☐ | For hierarchical and complex designs, identification of the appropriate level for tests and full reporting of outcomes |
| ☒ | ☐ | Estimates of effect sizes (e.g. Cohen's *d*, Pearson's *r*), indicating how they were calculated |

*Our web collection on statistics for biologists contains articles on many of the points above.*

## Software and code

Policy information about availability of computer code

| Data collection | No software used. |
|---|---|
| Data analysis | All correlated evolutionary analyses used BayesTraits version 3.0.5 (Pagel and Meade 2021). Maps throughout this paper were generated using ArcGIS Pro 3.5.3 and ESRI's World Terrain Reference. ArcGIS® and ArcMap™ are the intellectual property of Esri and are used herein under license. Copyright © Esri. All rights reserved. For more information about Esri® software, please visit www.esri.com. |

For manuscripts utilizing custom algorithms or software that are central to the research but not yet described in published literature, software must be made available to editors and reviewers. We strongly encourage code deposition in a community repository (e.g. GitHub). See the Nature Portfolio guidelines for submitting code & software for further information.

## Data

Policy information about availability of data

All manuscripts must include a data availability statement. This statement should provide the following information, where applicable:

- Accession codes, unique identifiers, or web links for publicly available datasets
- A description of any restrictions on data availability
- For clinical datasets or third party data, please ensure that the statement adheres to our policy

Data taken from D-Place https://d-place.org/

## Research involving human participants, their data, or biological material

Policy information about studies with human participants or human data. See also policy information about sex, gender (identity/presentation), and sexual orientation and race, ethnicity and racism.

| | |
|---|---|
| Reporting on sex and gender | *Use the terms sex (biological attribute) and gender (shaped by social and cultural circumstances) carefully in order to avoid confusing both terms. Indicate if findings apply to only one sex or gender; describe whether sex and gender were considered in study design; whether sex and/or gender was determined based on self-reporting or assigned and methods used.*<br>*Provide in the source data disaggregated sex and gender data, where this information has been collected, and if consent has been obtained for sharing of individual-level data; provide overall numbers in this Reporting Summary. Please state if this information has not been collected.*<br>*Report sex- and gender-based analyses where performed, justify reasons for lack of sex- and gender-based analysis.* |
| Reporting on race, ethnicity, or other socially relevant groupings | *Please specify the socially constructed or socially relevant categorization variable(s) used in your manuscript and explain why they were used. Please note that such variables should not be used as proxies for other socially constructed/relevant variables (for example, race or ethnicity should not be used as a proxy for socioeconomic status).*<br>*Provide clear definitions of the relevant terms used, how they were provided (by the participants/respondents, the researchers, or third parties), and the method(s) used to classify people into the different categories (e.g. self-report, census or administrative data, social media data, etc.)*<br>*Please provide details about how you controlled for confounding variables in your analyses.* |
| Population characteristics | *Describe the covariate-relevant population characteristics of the human research participants (e.g. age, genotypic information, past and current diagnosis and treatment categories). If you filled out the behavioural & social sciences study design questions and have nothing to add here, write "See above."* |
| Recruitment | *Describe how participants were recruited. Outline any potential self-selection bias or other biases that may be present and how these are likely to impact results.* |
| Ethics oversight | *Identify the organization(s) that approved the study protocol.* |

Note that full information on the approval of the study protocol must also be provided in the manuscript.

# Field-specific reporting

Please select the one below that is the best fit for your research. If you are not sure, read the appropriate sections before making your selection.

☐ Life sciences    ☒ Behavioural & social sciences    ☐ Ecological, evolutionary & environmental sciences

For a reference copy of the document with all sections, see nature.com/documents/nr-reporting-summary-flat.pdf

# Behavioural & social sciences study design

All studies must disclose on these points even when the disclosure is negative.

| | |
|---|---|
| Study description | This study matched 868 societies from the Ethnographic Atlas (EA) and the full 186 societies from the Standard Cross-Cultural Sample (SCCS) to the posterior treeset of the Global Supertree developed by Bouckaert and colleagues (Bouckaert, et al. 2022). Then quantitative data were used from both samples for traits covering agriculture, political complexity, tax and writing. Bayesian phylogenetic methods were then used to analyse those data. |
| Research sample | Data were collected for the EA and SCCS from D-Place, an online public database (https://d-place.org/). |
| Sampling strategy | The maximum sample size was used for each analysis - 868 from EA and 186 from SCCS. All analyses that did not include data on taxation or writing used EA data. |
| Data collection | Only data from public database D-Place were used. |
| Timing | Data downloaded September 2022. |
| Data exclusions | No data were excluded. |
| Non-participation | Participation not applicable. |
| Randomization | Randomization not applicable. |

# Reporting for specific materials, systems and methods

We require information from authors about some types of materials, experimental systems and methods used in many studies. Here, indicate whether each material, system or method listed is relevant to your study. If you are not sure if a list item applies to your research, read the appropriate section before selecting a response.

## Materials & experimental systems

| n/a | Involved in the study |
|---|---|
| ☐ ☐ | Antibodies |
| ☐ ☐ | Eukaryotic cell lines |
| ☐ ☐ | Palaeontology and archaeology |
| ☐ ☐ | Animals and other organisms |
| ☐ ☐ | Clinical data |
| ☐ ☐ | Dual use research of concern |
| ☐ ☐ | Plants |

## Methods

| n/a | Involved in the study |
|---|---|
| ☐ ☐ | ChIP-seq |
| ☐ ☐ | Flow cytometry |
| ☐ ☐ | MRI-based neuroimaging |

# Antibodies

| Antibodies used | Describe all antibodies used in the study; as applicable, provide supplier name, catalog number, clone name, and lot number. |
|---|---|
| Validation | Describe the validation of each primary antibody for the species and application, noting any validation statements on the manufacturer's website, relevant citations, antibody profiles in online databases, or data provided in the manuscript. |

# Eukaryotic cell lines

Policy information about cell lines and Sex and Gender in Research

| Cell line source(s) | State the source of each cell line used and the sex of all primary cell lines and cells derived from human participants or vertebrate models. |
|---|---|
| Authentication | Describe the authentication procedures for each cell line used OR declare that none of the cell lines used were authenticated. |
| Mycoplasma contamination | Confirm that all cell lines tested negative for mycoplasma contamination OR describe the results of the testing for mycoplasma contamination OR declare that the cell lines were not tested for mycoplasma contamination. |
| Commonly misidentified lines (See ICLAC register) | Name any commonly misidentified cell lines used in the study and provide a rationale for their use. |

# Palaeontology and Archaeology

| Specimen provenance | Provide provenance information for specimens and describe permits that were obtained for the work (including the name of the issuing authority, the date of issue, and any identifying information). Permits should encompass collection and, where applicable, export. |
|---|---|
| Specimen deposition | Indicate where the specimens have been deposited to permit free access by other researchers. |
| Dating methods | If new dates are provided, describe how they were obtained (e.g. collection, storage, sample pretreatment and measurement), where they were obtained (i.e. lab name), the calibration program and the protocol for quality assurance OR state that no new dates are provided. |

☐ Tick this box to confirm that the raw and calibrated dates are available in the paper or in Supplementary Information.

| Ethics oversight | Identify the organization(s) that approved or provided guidance on the study protocol, OR state that no ethical approval or guidance was required and explain why not. |
|---|---|

Note that full information on the approval of the study protocol must also be provided in the manuscript.

# Animals and other research organisms

Policy information about studies involving animals; ARRIVE guidelines recommended for reporting animal research, and Sex and Gender in Research

| Laboratory animals | For laboratory animals, report species, strain and age OR state that the study did not involve laboratory animals. |
|---|---|
| Wild animals | Provide details on animals observed in or captured in the field; report species and age where possible. Describe how animals were caught and transported and what happened to captive animals after the study (if killed, explain why and describe method; if released, say where and when) OR state that the study did not involve wild animals. |

| Reporting on sex | *Indicate if findings apply to only one sex; describe whether sex was considered in study design, methods used for assigning sex. Provide data disaggregated for sex where this information has been collected in the source data as appropriate; provide overall numbers in this Reporting Summary. Please state if this information has not been collected. Report sex-based analyses where performed, justify reasons for lack of sex-based analysis.* |
|---|---|
| Field-collected samples | *For laboratory work with field-collected samples, describe all relevant parameters such as housing, maintenance, temperature, photoperiod and end-of-experiment protocol OR state that the study did not involve samples collected from the field.* |
| Ethics oversight | *Identify the organization(s) that approved or provided guidance on the study protocol, OR state that no ethical approval or guidance was required and explain why not.* |

Note that full information on the approval of the study protocol must also be provided in the manuscript.

# Clinical data

Policy information about clinical studies

All manuscripts should comply with the ICMJE guidelines for publication of clinical research and a completed CONSORT checklist must be included with all submissions.

| Clinical trial registration | *Provide the trial registration number from ClinicalTrials.gov or an equivalent agency.* |
|---|---|
| Study protocol | *Note where the full trial protocol can be accessed OR if not available, explain why.* |
| Data collection | *Describe the settings and locales of data collection, noting the time periods of recruitment and data collection.* |
| Outcomes | *Describe how you pre-defined primary and secondary outcome measures and how you assessed these measures.* |

# Dual use research of concern

Policy information about dual use research of concern

## Hazards

Could the accidental, deliberate or reckless misuse of agents or technologies generated in the work, or the application of information presented in the manuscript, pose a threat to:

No | Yes

☐ ☐ Public health

☐ ☐ National security

☐ ☐ Crops and/or livestock

☐ ☐ Ecosystems

☐ ☐ Any other significant area

## Experiments of concern

Does the work involve any of these experiments of concern:

No | Yes

☐ ☐ Demonstrate how to render a vaccine ineffective

☐ ☐ Confer resistance to therapeutically useful antibiotics or antiviral agents

☐ ☐ Enhance the virulence of a pathogen or render a nonpathogen virulent

☐ ☐ Increase transmissibility of a pathogen

☐ ☐ Alter the host range of a pathogen

☐ ☐ Enable evasion of diagnostic/detection modalities

☐ ☐ Enable the weaponization of a biological agent or toxin

☐ ☐ Any other potentially harmful combination of experiments and agents

# Plants

Seed stocks | *Report on the source of all seed stocks or other plant material used. If applicable, state the seed stock centre and catalogue number. If plant specimens were collected from the field, describe the collection location, date and sampling procedures.*

Novel plant genotypes | *Describe the methods by which all novel plant genotypes were produced. This includes those generated by transgenic approaches, gene editing, chemical/radiation-based mutagenesis and hybridization. For transgenic lines, describe the transformation method, the number of independent lines analyzed and the generation upon which experiments were performed. For gene-edited lines, describe the editor used, the endogenous sequence targeted for editing, the targeting guide RNA sequence (if applicable) and how the editor was applied.*

Authentication | *Describe any authentication procedures for each seed stock used or novel genotype generated. Describe any experiments used to assess the effect of a mutation and, where applicable, how potential secondary effects (e.g. second site T-DNA insertions, mosaicism, off-target gene editing) were examined.*

# ChIP-seq

## Data deposition

☐ Confirm that both raw and final processed data have been deposited in a public database such as GEO.

☐ Confirm that you have deposited or provided access to graph files (e.g. BED files) for the called peaks.

Data access links
*May remain private before publication.* | *For "Initial submission" or "Revised version" documents, provide reviewer access links. For your "Final submission" document, provide a link to the deposited data.*

Files in database submission | *Provide a list of all files available in the database submission.*

Genome browser session
(e.g. UCSC) | *Provide a link to an anonymized genome browser session for "Initial submission" and "Revised version" documents only, to enable peer review. Write "no longer applicable" for "Final submission" documents.*

## Methodology

Replicates | *Describe the experimental replicates, specifying number, type and replicate agreement.*

Sequencing depth | *Describe the sequencing depth for each experiment, providing the total number of reads, uniquely mapped reads, length of reads and whether they were paired- or single-end.*

Antibodies | *Describe the antibodies used for the ChIP-seq experiments; as applicable, provide supplier name, catalog number, clone name, and lot number.*

Peak calling parameters | *Specify the command line program and parameters used for read mapping and peak calling, including the ChIP, control and index files used.*

Data quality | *Describe the methods used to ensure data quality in full detail, including how many peaks are at FDR 5% and above 5-fold enrichment.*

Software | *Describe the software used to collect and analyze the ChIP-seq data. For custom code that has been deposited into a community repository, provide accession details.*

# Flow Cytometry

## Plots

Confirm that:

☐ The axis labels state the marker and fluorochrome used (e.g. CD4-FITC).

☐ The axis scales are clearly visible. Include numbers along axes only for bottom left plot of group (a 'group' is an analysis of identical markers).

☐ All plots are contour plots with outliers or pseudocolor plots.

☐ A numerical value for number of cells or percentage (with statistics) is provided.

## Methodology

Sample preparation | *Describe the sample preparation, detailing the biological source of the cells and any tissue processing steps used.*

Instrument | *Identify the instrument used for data collection, specifying make and model number.*

Software | *Describe the software used to collect and analyze the flow cytometry data. For custom code that has been deposited into a community repository, provide accession details.*

| Cell population abundance | *Describe the abundance of the relevant cell populations within post-sort fractions, providing details on the purity of the samples and how it was determined.* |
| Gating strategy | *Describe the gating strategy used for all relevant experiments, specifying the preliminary FSC/SSC gates of the starting cell population, indicating where boundaries between "positive" and "negative" staining cell populations are defined.* |

☐ Tick this box to confirm that a figure exemplifying the gating strategy is provided in the Supplementary Information.

# Magnetic resonance imaging

## Experimental design

| Design type | *Indicate task or resting state; event-related or block design.* |
| Design specifications | *Specify the number of blocks, trials or experimental units per session and/or subject, and specify the length of each trial or block (if trials are blocked) and interval between trials.* |
| Behavioral performance measures | *State number and/or type of variables recorded (e.g. correct button press, response time) and what statistics were used to establish that the subjects were performing the task as expected (e.g. mean, range, and/or standard deviation across subjects).* |

## Acquisition

| Imaging type(s) | *Specify: functional, structural, diffusion, perfusion.* |
| Field strength | *Specify in Tesla* |
| Sequence & imaging parameters | *Specify the pulse sequence type (gradient echo, spin echo, etc.), imaging type (EPI, spiral, etc.), field of view, matrix size, slice thickness, orientation and TE/TR/flip angle.* |
| Area of acquisition | *State whether a whole brain scan was used OR define the area of acquisition, describing how the region was determined.* |

Diffusion MRI ☐ Used ☐ Not used

## Preprocessing

| Preprocessing software | *Provide detail on software version and revision number and on specific parameters (model/functions, brain extraction, segmentation, smoothing kernel size, etc.).* |
| Normalization | *If data were normalized/standardized, describe the approach(es): specify linear or non-linear and define image types used for transformation OR indicate that data were not normalized and explain rationale for lack of normalization.* |
| Normalization template | *Describe the template used for normalization/transformation, specifying subject space or group standardized space (e.g. original Talairach, MNI305, ICBM152) OR indicate that the data were not normalized.* |
| Noise and artifact removal | *Describe your procedure(s) for artifact and structured noise removal, specifying motion parameters, tissue signals and physiological signals (heart rate, respiration).* |
| Volume censoring | *Define your software and/or method and criteria for volume censoring, and state the extent of such censoring.* |

## Statistical modeling & inference

| Model type and settings | *Specify type (mass univariate, multivariate, RSA, predictive, etc.) and describe essential details of the model at the first and second levels (e.g. fixed, random or mixed effects; drift or auto-correlation).* |
| Effect(s) tested | *Define precise effect in terms of the task or stimulus conditions instead of psychological concepts and indicate whether ANOVA or factorial designs were used.* |

Specify type of analysis: ☐ Whole brain ☐ ROI-based ☐ Both

| Statistic type for inference<br><br>(See Eklund et al. 2016) | *Specify voxel-wise or cluster-wise and report all relevant parameters for cluster-wise methods.* |
| Correction | *Describe the type of correction and how it is obtained for multiple comparisons (e.g. FWE, FDR, permutation or Monte Carlo).* |

## Models & analysis

| n/a | Involved in the study |
|-----|----------------------|
| ☐ ☐ | Functional and/or effective connectivity |
| ☐ ☐ | Graph analysis |
| ☐ ☐ | Multivariate modeling or predictive analysis |

**Functional and/or effective connectivity**

*Report the measures of dependence used and the model details (e.g. Pearson correlation, partial correlation, mutual information).*

**Graph analysis**

*Report the dependent variable and connectivity measure, specifying weighted graph or binarized graph, subject- or group-level, and the global and/or node summaries used (e.g. clustering coefficient, efficiency, etc.).*

**Multivariate modeling and predictive analysis**

*Specify independent variables, features extraction and dimension reduction, model, training and evaluation metrics.*

