## [Peer Review File · Nature Human Behaviour]

State formation across cultures and the role of grain, intensive agriculture, taxation and writing.

Corresponding Author: Dr Christopher Opie

Version 0:

Decision Letter:

23rd February 2024

Dear Dr Opie,

Thank you once again for your manuscript, entitled "The origin of the state: grain, tax and writing", and for your patience during the peer review process.

Your Article has now been evaluated by 3 referees. You will see from their comments copied below that, although they find your work of potential interest, they have raised quite substantial concerns. In light of these comments, we cannot accept the manuscript for publication, but would be interested in considering a revised version if you are willing and able to fully address reviewer and editorial concerns.

We hope you will find the referees' comments useful as you decide how to proceed. If you wish to submit a substantially revised manuscript, please bear in mind that we will be reluctant to approach the referees again in the absence of major revisions. We are committed to providing a fair and constructive peer-review process. Do not hesitate to contact us if there are specific requests from the reviewers that you believe are technically impossible or unlikely to yield a meaningful outcome.

If you wish to submit a suitably revised manuscript, we would hope to receive it within 4 months. I would be grateful if you could contact us as soon as possible if you foresee difficulties with meeting this target resubmission date.

- Include a "Response to the editors and reviewers" document detailing, point-by-point, how you addressed each editor and referee comment. If no action was taken to address a point, you must provide a compelling argument. When formatting this document, please respond to each reviewer comment individually, including the full text of the reviewer comment verbatim followed by your response to the individual point. This response will be used by the editors to evaluate your revision and sent back to the reviewers along with the revised manuscript.
- Highlight all changes made to your manuscript or provide us with a version that tracks changes.

Link Redacted

Thank you for the opportunity to review your work. Please do not hesitate to contact me if you have any questions or would like to discuss the required revisions further.

Sincerely,

[Redacted signature block]

Reviewer expertise:

Reviewer #1: Agriculture and state formation; EA and SCCS

Reviewer #2: Agriculture and state formation; EA and SCCS

Reviewer #3: Bayesian phylogenetic analyses

REVIEWER COMMENTS:

Reviewer #1:

Remarks to the Author:

The authors of this manuscript use data from the 186 societies of the Standard Cross Cultural Sample, collected by anthropologists during the 1960s and 1970s, to assess several hypotheses about writing, agriculture, and the development of the state. The key potential advance in the current manuscript compared to previous work is to use phylogenetic analysis that may allow for establishing the temporal relationship between the emergence of agriculture, writing, and the state.

The background here is that in recent years a number of social scientists, especially economists and political scientists, have used this data source to show first that states developed in areas that were suitable for cereal based agriculture but not in areas suitable for agriculture based on roots and tubers. The inference drawn for this conclusion is that because cereals can be stored this would allow for the production of a surplus that could be used for a class of individuals who would not be directly engaged in agriculture. In contrast, roots and tubers generally cannot be stored. Recently, further work has added to this, using the same dataset, to highlight the relationship between the development of writing and the origin of the state.

A key limitation of the Standard Cross Cultural Sample dataset is that it is strictly cross-sectional, which makes it difficult to establish temporal relationships between variables. We can use the SCCS to establish whether a society had agriculture, writing, and a state, but we cannot use it to establish which of these elements came first. This is where the main contribution of the current manuscript comes in. The authors use a phylogenetic analysis of languages from Bouckart et al. (2022) to establish inferences about the relationship between the different societies in the SCCS. It would appear that this could then be used to establish inferences about the likely temporal relationship between agriculture, writing, and the state.

Unfortunately, this is about as far as I made it with understanding the relative contribution of this paper. The method of Bouckart is not really explained here. Nor is there any real discussion of the assumptions necessary for the application of this method. I presume that the evidence from Bouckart et al. allow us to which societies borrowed elements of their language from other societies and this may also allow us to draw inferences about the origins of other sociocultural characteristics? It is possible that I am missing something obvious, and if so, my review should be discounted. Otherwise, it seems like this manuscript might be better suited for a specialized outlet where readers will be certain to be familiar with the methods used, as opposed to a general interest journal like Nature Human Behaviour.

It is also the case that the empirical results here are presented rather abruptly and without much explanation.

Reviewer #2:

None

Reviewer #3:

Remarks to the Author:

This paper tackles an interesting and important topic – the role of intensive, grain agriculture, taxation and writing in the formation of states. It does so using a new global language phylogeny (a “supertree”), ethnographic data (868 and 186 societies), and a series of tests for correlated evolution.

Based on the topic, it is well worth publishing in Nature Human Behaviour. The supertree is not the first of its kind, but it is nonetheless a new tool in evolutionary anthropology that may be used for similar global phylogenetic cross-cultural analyses in the future. Unfortunately, the paper suffers from inadequate description of methods, poor presentation of data, and poor writing (especially in the discussion). Crucially, the authors misinterpret the results of the test of correlated evolution. This makes the paper unsuitable for publication in any journal, in its current form.

The Introduction section is brief and straightforward. I would consider it sufficient if it were made up for by a longer, more thorough discussion, but that is not the case. The introduction would benefit from delving into what social complexity actually is. For example, a recent paper (a preprint) by Martin et al. (“Novel phylogenetic methods reveal that resource-use intensification drives the evolution of “complex” societies”) tackles a similar issue using similar data and methods. It conceptualizes social complexity as a set of related features (population density, social stratification, political integration...) that relate to subsistence. Here, a single variable (EA033/SCCS237 - jurisdictional hierarchy) is used without explanation or simply saying what defines a state (what is the difference between state and non-state), what precedes state organization, etc.

The Methods section only describes the variables that were used and the SI adds information on how these variables were coded, but the readers are not given the information about the distribution of character states (geographic or phylogenetic). The readers

should have some sense of where the societies in question are, or what proportion of them is organized into states, have grain agriculture, etc. (adding some numbers in Supplementary Table 1 is a bare minimum). Another question is how much information there is about the agriculture and jurisdictional hierarchy in the EA, compared to taxation and writing in the SCCS, because there is a big difference in the sample size.

The paper boasts of a new phylogeny, but does not show it. A maximum credibility tree should be shown, ideally with the distribution of character states in question - social complexity, etc., both the EA and the SCCS subset tree (at least in the SI). There are other supertrees available (Minocher et al. 2019; Saffa et al., 2022), but the readers cannot tell how is the new tree different. The tree has, in fact, not been published in a journal, it was only presented in a preprint that was posted online about 2 years ago. That preprint also does not show a maximum credibility tree, only a densitree, which does not allow to assess whether the topology or chronology of the tree makes sense. (The description of the tree raises questions. Why, for example, is there highest phylogenetic diversity in Oceania, when it is not is not linguistically most diverse region and it has been colonized relatively recently?) Importantly, the supertree has not been made publicly available. Another question: The "supertree" includes over 90% of worldwide linguistic diversity, but the subtree utilized here only includes about 2/3 of the societies in the EA. Why? What is the reason for this limited overlap?

The paper relies on a single phylogenetic comparative analysis: a Bayesian implementation of Pagel's test of correlated evolution. The results of this analysis are quite difficult to interpret and the Results section is a chore to read through. It does not help that the transition rate plots (Figure 1–6) are the only figures in the paper. Crucially, the authors themselves have the tendency to misinterpret the results of this analysis. I will explain using the first result as an example:

Line 121: "The rate matrix in Figure 1 indicates that the presence of intensive agriculture makes the emergence of states somewhat more likely – the rate of transition to statehood in the presence of agriculture (q34) is twice the rate of transition to statehood in the absence of agriculture (q12). However, we find much stronger evidence that the presence of a state makes the transition to intensive agriculture more likely (q24 is six times the rate of q13)." This is the principle finding of the paper. Line 11 in the abstract reads: "Our results suggest that intensive agriculture was as likely the result of state formation as its cause."

One cannot read the results of Pagel's test like this. The test indicates how "stable" different combination of character states are on a phylogeny (given their distribution, the tree topology, branch lengths, and the model). The test only tells you what tends to happen if... If, for example, the statehood and the absence of agriculture (a minority case, cross-culturally) find themselves in the same node of the tree. If statehood somehow emerges without intensive agriculture, the rate matrix (Figure 1) indicates that this is evolutionarily unstable state and there is the tendency to "run away" to more stable state, that is, a combination of statehood and intensive agriculture (q24 in Figure 1). The authors interpret this high transition rate as evidence for formation of state preceding the intensification of agriculture, but it is, in fact, the other way around. This is why the results are (seemingly) at odds with other studies, including those cited in the introduction (or the aforementioned Martin et al. paper).

Similarly, that "non-grain agriculture was much more likely to be lost in states than in non-states" (line 166) does not actually suggests that "states encouraged the growing of grain at the expense of other forms of agriculture" (line 167). Pagel's test shows that the combination of statehood and non-grain agriculture is unstable, but says nothing about the direction of causality. Rather, agriculture (that is an intensive, grain agriculture), comes first, state comes later and then, grain agriculture, never loses its importance (or primacy). There is, most likely, no need to encourage or discourage production of roots or tubers. Is there some evidence to suggest otherwise? Because Pagel's test really can't decide this. That is not to say that authors interpret every single transition rate matrix incorrectly, but they have a tendency to interpret the results of Pagel's test causally, which makes the paper severely flawed.

The discussion is short and inadequate and does little more than just reiterating the results in the first three paragraphs. The last, fourth paragraph reads like a short advocacy for using phylogenetic comparative methods in anthropology and has little to do with the actual topic of the paper. The discussion, to put it bluntly, lacks discussion. There is no confrontation with the vast literature on the subject, not even with the few studies that are cited in the introduction. The authors gloss over the fact that their results contradict those of other authors (intensification of agriculture first, social complexity later), or discuss what „intensification“, or "complexity" are and how the way they are operationalized can affect the results. Since the results of the Pagel's test are conditional on the distribution of the combination of character states at the tips of the phylogeny, it may be instructive to look at minority cases that can help explain the general pattern. What are, for example, those states that rely on non-grain agriculture? How does taxation work in the absence of writing and what level of social complexity does it allow? Is writing ever lost? ... Last, but not least, it would be nice to acknowledge the role of environmental constraints in formation of states. The authors make it seem like it is easy to switch from non-grain to grain agriculture once social complexity increases and with it the need for crops with taxable potential, but that is not the case, as ideal taxable crops may simply not be available. The paper does not discuss that either.

In summary, while I find the topic of the paper interesting, I cannot recommend it for publication in Nature Human Behaviour.

Version 1:

Decision Letter:

18th December 2024

Dear Dr Opie,

Thank you once again for your manuscript, entitled "The origin of the state: grain, tax and writing," and for your patience during the peer review process.

Your manuscript has now been evaluated by 3 reviewers. Reviewers 1 and 2 did not include comments to authors, but in their comments to editors, Reviewer 1 indicated that they are happy with the revised manuscript, while Reviewer 2 raised significant concerns about the extent to which the paper is accessible (and therefore valuable) to a broad audience. Reviewer 3's comments are included at the end of this letter.

Overall, while the reviewers continue to find your work to be of interest, they also raise some important concerns. We remain interested in the possibility of publishing your study in Nature Human Behaviour, but would like to consider your response to these concerns in the form of a revised manuscript before we make a decision on publication.

Editorially we ask that you prioritise the following points:

1. It is crucial that the current work can stand alone, and for this additional methodological information is required. Please provide this, and reference a preprint of the article currently in review at Nature, and also studies that have used the tree within the main text. If the under-review paper at Nature has in the meantime been accepted for publication, please update the text to reflect this.
2. It is also crucial that the text (including the methods) is accessible to a broad multidisciplinary audience. Please improve the main text accordingly, in order to maximize the use-value and impact of your work for our multi-disciplinary readership.

In sum, we invite you to revise your manuscript taking into account all reviewer and editor comments. We are committed to providing a fair and constructive peer-review process. Do not hesitate to contact us if there are specific requests from the reviewers that you believe are technically impossible or unlikely to yield a meaningful outcome.

We hope to receive your revised manuscript within two months. I would be grateful if you could contact us as soon as possible if you foresee difficulties with meeting this target resubmission date.

- Include a "Response to the editors and reviewers" document detailing, point-by-point, how you addressed each editor and referee comment. If no action was taken to address a point, you must provide a compelling argument. When formatting this document, please respond to each reviewer comment individually, including the full text of the reviewer comment verbatim followed by your response to the individual point. This response will be used by the editors to evaluate your revision and sent back to the reviewers along with the revised manuscript.
- Highlight all changes made to your manuscript or provide us with a version that tracks changes.

Link Redacted

We look forward to seeing the revised manuscript and thank you for the opportunity to review your work. Please do not hesitate to contact me if you have any questions or would like to discuss these revisions further.

Sincerely,

[Redacted]

[Redacted]

Nature Human Behaviour

REVIEWER COMMENTS:

Reviewer #3 (Remarks to the Author):

I was very critical of this manuscript in my initial review, and I commend the authors for their thoughtful and rigorous response. The manuscript has been greatly improved.

The main changes in this version include a more comprehensive explanation of the methods and results, particularly the selection and coding of the key variable (social complexity) and the statistical methods used, along with clearer interpretations. New figures have been added, and the discussion has been completely rewritten. Several smaller changes further enhance the manuscript and make the presentation of the results more compelling.

I understand that a detailed description of the phylogenetic inference methods is available in an unpublished paper. The inclusion of the phylogenetic and geographic distribution of the studied variables is a significant improvement. The coevolution analysis is now described in a way that is more accessible to uninitiated readers, and the paper no longer relies so heavily on describing transition rate matrices, which made the previous version somewhat difficult to read.

The separate analysis for Niger-Congo languages in sub-Saharan Africa is a welcome addition. While it may be evident that grain is not a driver of state formation in this region (given the relative scarcity of complex societies there), including this analysis adds nuance and demonstrates attention to regional variation.

The rewritten discussion is well-crafted, and I have little to criticize. It might benefit from eliminating some repetitive statements (e.g., references to accounting for phylogenetic non-independence). A large dataset and a new supertree aside, this is a simple, elegant study and a discussion can be short.

I have a few minor comments and recommendations:

- Lines 51–54: This text seems misplaced. It addresses a methodological issue that would be better suited to the end of the introduction rather than its beginning. Most of the paragraph (lines 45–54) might fit better on page 2 when discussing the conceptualization of social complexity.
- Line 267: Citing R Development Core Team (2013) seems unnecessary in this context. If R was specifically used to create the Niger-Congo subtree, the sentence should be rephrased to clarify this. Alternatively, it may be better to remove the citation altogether.
- I find the term "cultural phylogeny" used for the supertree somewhat misleading. The tree is based on lexical data and a set of topological constraints based on genetics and the archaeological record, correct? I assume the authors want to emphasize that this supertree is well-suited for capturing cultural macroevolution (of social complexity, music, numerals, etc.), but wouldn't "language phylogeny" be more appropriate? After all, "cultural phylogenies" can also be constructed from manuscripts, oral traditions, artifacts, etc.
- The maps in the supplementary materials are of low resolution and difficult to read.
- Transition rate matrices based on the SCCS, which were included in the main text of the previous version, should be added to the supplementary materials for completeness.

Nitpicking aside, I believe this manuscript is now a strong contribution to the field, and I can recommend it for publication in Nature Human Behaviour.

Version 2:

Decision Letter:

Our ref: NATHUMBEHAV-23113961B

5th June 2025

Dear Dr. Opie,

Thank you for submitting your revised manuscript "The origin of the state: grain, tax and writing" (NATHUMBEHAV-23113961B). It has now been seen by Reviewer 3 from the previous rounds, and although they did not include any comments to authors, in their confidential comments to editors they noted that the paper has significantly improved in revision. We will therefore be happy in principle to publish it in Nature Human Behaviour, pending minor revisions to comply with our editorial and formatting guidelines.

We are now performing detailed checks on your paper and will send you a checklist detailing our editorial and formatting requirements within two weeks. Please do not upload the final materials and make any revisions until you receive this additional information from us.

Sincerely,

██████████
██████████
██████████
Nature Human Behaviour

Version 3:

Decision Letter:

Dear Dr Opie,

We are pleased to inform you that your Article "State formation across cultures and the role of grain, intensive agriculture, taxation and writing.", has now been accepted for publication in Nature Human Behaviour.

Authors may need to take specific actions to achieve compliance with funder and institutional open access mandates. If your research is supported by a funder that requires immediate open access (e.g. according to [Plan S principles](https://www.springernature.com/gp/open-science/plan-s-compliance) or the [NIH public access policy](https://www.springernature.com/gp/open-science/us-federal-agency-compliance)) then you should select the gold OA route, and we will direct you to the compliant route where possible. Because authors warrant under our subscription licensing terms that they haven't committed to licensing any version of their article under a licence inconsistent with the terms of our agreement – including the applicable embargo period – publication under the subscription model isn't suitable for authors whose funders require no embargo.

With best regards,

██████████

P.S. Click on the following link if you would like to recommend Nature Human Behaviour to your librarian
<http://www.nature.com/subscriptions/recommend.html#forms>

** Visit the Springer Nature Editorial and Publishing website at http://editorial-jobs.springernature.com?utm_source=ejp_NHumB_email&utm_medium=ejp_NHumB_email&utm_campaign=ejp_NHumB for more information about our career opportunities. If you have any questions please click [here](mailto:editorial.publishing.jobs@springernature.com).

Open Access This Peer Review File is licensed under a Creative Commons Attribution 4.0 International License, which permits use, sharing, adaptation, distribution and reproduction in any medium or format, as long as you give appropriate credit to the original author(s) and the source, provide a link to the Creative Commons license, and indicate if changes were made. In cases where reviewers are anonymous, credit should be given to 'Anonymous Referee' and the source. The images or other third party material in this Peer Review File are included in the article's Creative Commons license, unless indicated otherwise in a credit line to the material. If material is not included in the article's Creative Commons license and your intended use is not permitted by statutory regulation or exceeds the permitted use, you will need to obtain permission directly from the copyright holder.

Response to the editors and reviewers

Reviewer #1:

Remarks to the Author:

The authors of this manuscript use data from the 186 societies of the Standard Cross Cultural Sample, collected by anthropologists during the 1960s and 1970s, to assess several hypotheses about writing, agriculture, and the development of the state. The key potential advance in the current manuscript compared to previous work is to use phylogenetic analysis that may allow for establishing the temporal relationship between the emergence of agriculture, writing, and the state.

The background here is that in recent years a number of social scientists, especially economists and political scientists, have used this data source to show first that states developed in areas that were suitable for cereal based agriculture but not in areas suitable for agriculture based on roots and tubers. The inference drawn for this conclusion is that because cereals can be stored this would allow for the production of a surplus that could be used for a class of individuals who would not be directly engaged in agriculture. In contrast, roots and tubers generally cannot be stored. Recently, further work has added to this, using the same dataset, to highlight the relationship between the development of writing and the origin of the state.

A key limitation of the Standard Cross Cultural Sample dataset is that it is strictly cross-sectional, which makes it difficult to establish temporal relationships between variables. We can use the SCCS to establish whether a society had agriculture, writing, and a state, but we cannot use it to establish which of these elements came first. This is where the main contribution of the current manuscript comes in. The authors use a phylogenetic analysis of languages from Bouckart et al. (2022) to establish inferences about the relationship between the different societies in the SCCS. It would appear that this could then be used to establish inferences about the likely temporal relationship between agriculture, writing, and the state.

Unfortunately, this is about as far as I made it with understanding the relative contribution of this paper. The method of Bouckart is not really explained here. Nor is there any real discussion of the assumptions necessary for the application of this method. I presume that the evidence from Bouckart et al. allow us to which societies borrowed elements of their language from other societies and this may also allow us to draw inferences about the origins of other sociocultural characteristics? It is possible that I am missing something obvious, and if so, my review should be discounted. Otherwise, it seems like this manuscript might be better suited for a specialized outlet where readers will be certain to be familiar with the methods used, as opposed to a general interest journal like Nature Human Behaviour.

Thank you for the useful comments, which we hope we have responded to. We have expanded the explanation (in Results, Discussion and Methods) for the way phylogenetic methods using the new global language tree and the Bayesian Discrete

approach can be used to establish correlated evolution between binary traits taking account of the uncertainty within the global phylogeny. We have also expanded our explanation of how the Bayesian phylogenetic approach can establish the relative timing for the evolution of the traits in question, and discuss the assumptions of the method. This general approach, initially developed in evolutionary biology, has been used by many studies investigating the co-evolution of cultural traits, including in the pages of general interest journals like *Nature* (e.g. Currie, et al. 2010; Dunn, et al. 2011; Watts, et al. 2016) and *Nature Human Behaviour* (e.g. Basava, et al. 2021; Šaffa, et al. 2022; Sheehan, et al. 2022). The global phylogeny we use extends the Bayesian phylogenetic inference approach developed by Atkinson and colleagues (see, for example, publications in *Science* and *Nature Ecology and Evolution* Bouckaert, et al. 2018; Bouckaert, et al. 2012; Heggarty, et al. 2023) to a global scale. The global tree is published as a preprint and is currently under revision with *Nature*, but the posterior distribution of trees in the preprint have already been used in several publications including in *Nature Communications* (Passmore, et al. 2024), *Science Advances* (Skirgård, et al. 2023), and *Humanities and Social Sciences Communications* (Her, et al. 2024). Ours is the first paper to combine this newly available global phylogeny with the Bayesian Discrete approach to explore the development of the state worldwide.

It is also the case that the empirical results here are presented rather abruptly and without much explanation.

We have rewritten the results section in light of these comments and hope that it now expands on the explanation of the results.

Reviewer #2:

Remarks to the Author:

None

Reviewer #3:

Remarks to the Author:

This paper tackles an interesting and important topic – the role a role of intensive, grain agriculture, taxation and writing in the formation of states. It does so using a new global language phylogeny (a “supertree”), ethnographic data (868 and 186 societies), and a series of tests for correlated evolution.

Based on the topic, it is well worth publishing in Nature Human Behaviour. The supertree is not the first of its kind, but it is nonetheless a new tool in evolutionary anthropology that may be used for similar global phylogenetic cross-cultural analyses in the future. Unfortunately, the paper suffers from inadequate description of methods, poor presentation of data, and poor writing (especially in the discussion). Crucially, the authors misinterpret the results of the test of correlated evolution. This makes the paper unsuitable for publication in any journal, in its current form.

Thank you for the comprehensive review of our paper. We hope that we have been able to respond to your comments below and that it is now suitable for publication in *Nature Human Behaviour*.

The Introduction section is brief and straightforward. I would consider it sufficient if it were made up for by a longer, more thorough discussion, but that is not the case. The introduction would benefit from delving into what social complexity actually is. For example, a recent paper (a preprint) by Martin et al. (“Novel phylogenetic methods reveal that resource-use intensification drives the evolution of “complex” societies”) tackles a similar issue using similar data and methods. It conceptualizes social complexity as a set of related features (population density, social stratification, political integration...) that relate to subsistence. Here, a single variable (EA033/SCCS237 – jurisdictional hierarchy) is used without explanation or simply saying what defines a state (what is the difference between state and non-state), what precedes state organization, etc.

The paper by Martin et al. referred to here uses dimension reduction techniques to map the relationship between resource-use intensification and a cluster of traits that co-occur in “complex” societies. While an interesting question in its own right, the dimension reduction employed means the findings cannot speak to how resource intensification relates to specific elements of, or thresholds in, complexity, which is the focus of our study. We now explain that we have used a single variable (Jurisdictional Hierarchy) because this variable relates specifically to the decades of prior theoretical and empirical work on the transition from pre-state societies to states and the hypotheses that we are testing relate specifically to the origin of the state. This variable has been widely used for this purpose in previous studies (e.g. Currie, et al. 2010; Currie and Mace 2009; Mayshar, et al. 2022; Michalopoulos and Papaioannou 2013; Stasavage 2021). We have now given a clear explanation of the levels of Jurisdictional Hierarchy for a state, the differences between states and non-states and the characteristics of non-states (lines 49-54, 144-145, 640-643).

The Methods section only describes the variables that were used and the SI adds information on how these variables were coded, but the readers are not given the information about the distribution of character states (geographic or phylogenetic). The readers should have some sense of where the societies in question are, or what proportion of them is organized into states, have grain agriculture, etc. (adding some numbers in Supplementary Table 1 is a bare minimum). Another question is how much information there is about the agriculture and jurisdictional hierarchy in the EA, compared to taxation and writing in the SCCS, because there is a big difference in the sample size.

We have now added figures for the phylogenetic distribution of each of the variables from the Ethnographic Atlas and the Standard Cross-Cultural Sample. We have included Jurisdictional Hierarchy, Intensive Agriculture, and crop type (Figure 1) in the paper itself as the variables being investigated using EA data, and put the figure for the SCCS variables in the SI (Figure S1). SI Table 1 now includes the totals for each of the variables. Each of the results now includes crosstabs for the trait pairs. We have also included the geographic distribution of traits. State and grain data are included in the manuscript (Figures 2 & 3), and other traits are shown in the SI (Figures S2-S5).

The paper boasts of a new phylogeny, but does not show it. A maximum credibility tree should be shown, ideally with the distribution of character states in question - social complexity, etc., both the EA and the SCCS subset tree (at least in the SI).

There are other supertrees available (Minocher et al. 2019; Šaffa et al., 2022), but the readers cannot tell how is the new tree different. The tree has, in fact, not been published in a journal, it was only presented in a preprint that was posted online about 2 years ago. That preprint also does not show a maximum credibility tree, only a densitree, which does not allow to assess whether the topology or chronology of the tree makes sense. (The description of the tree raises questions. Why, for example, is there highest phylogenetic diversity in Oceania, when it is not is not linguistically most diverse region and it has been colonized relatively recently?) Importantly, the supertree has not been made publicly available. Another question: The “supertree” includes over 90% of worldwide linguistic diversity, but the subtree utilized here only includes about 2/3 of the societies in the EA. Why? What is the reason for this limited overlap?

A major advantage of the (Bouckaert, et al. 2022) treeset is precisely that it does not emphasise a single tree, but rather a posterior distribution of 1000 trees that represent the considerable phylogenetic uncertainty in relationships between the world’s languages. This is why the main figure in the paper is a densitree (emphasising the uncertainty) rather than an MCC tree (emphasising a single tree from the posterior). Crucially, by making inferences across the posterior distribution of trees, it is possible to incorporate phylogenetic uncertainty into any inferences, rather than assuming a particular ‘best’ tree topology. Having said this, we totally agree that an MCC tree can be useful to visualise what one tree looks like, and how data maps onto that tree. We therefore now include the MCCT of the phylogeny (Bouckaert, et al. 2022) with each of the variables (EA and SCCS) in the paper for Jurisdictional Hierarchy, Intensive Agriculture and Grain, and the SI for the other variables. We explain the benefits of using the Bouckaert et al. phylogeny. This treeset has been published as a preprint and is currently in review at Nature. The posterior distribution and all code used to generate it is also publicly available (GitHub link) and has been used in 3 peer-reviewed studies in the journals *Science Advances* (Skirgård, et al. 2023), *Nature Communications* (Passmore, et al. 2024), and *Humanities and Social Sciences Communications* (Her, et al. 2024).

The global treeset we use includes all or almost all extant languages identified via glottocodes. These languages have been matched to the Ethnographic Atlas (EA) societies using D-Place (Kirby, et al. 2016). Not all EA societies are in the global tree since some EA societies are extinct. The language used by each EA society has been checked using Glottolog (<https://glottolog.org/>). Only if the language name on the phylogeny is the same as the society name, or Glottolog clearly identifies that the language is spoken by that society, is the society included. This is a conservative approach that ensures that the EA societies used are accurately matched to the phylogeny.

The paper relies on a single phylogenetic comparative analysis: a Bayesian implementation of Pagel’s test of correlated evolution. The results of this analysis are quite difficult to interpret and the Results section is a chore to read through. It does not help that the transition rate plots (Figure 1–6) are the only figures in the paper. Crucially, the authors themselves have the tendency to misinterpret the results of this analysis. I will explain using the first result as an example:

We use Discrete (part of BayesTraits) (Pagel and Meade 2006) the leading method used across evolutionary biology and anthropology for detecting correlated evolution between discrete traits (cited by ~ 1000 studies). The limitation of this method, it has been argued, is that it requires traits to be binary. In our case however, the hypotheses we are testing are appropriately identified as binary, so we don't believe that is a problem.

Line 121: "The rate matrix in Figure 1 indicates that the presence of intensive agriculture makes the emergence of states somewhat more likely – the rate of transition to statehood in the presence of agriculture (q34) is twice the rate of transition to statehood in the absence of agriculture (q12). However, we find much stronger evidence that the presence of a state makes the transition to intensive agriculture more likely (q24 is six times the rate of q13)." This is the principle finding of the paper. Line 11 in the abstract reads: "Our results suggest that intensive agriculture was as likely the result of state formation as its cause."

One cannot read the results of Pagel's test like this. The test indicates how "stable" different combination of character states are on a phylogeny (given their distribution, the tree topology, branch lengths, and the model). The test only tells you what tends to happen if... If, for example, the statehood and the absence of agriculture (a minority case, cross-culturally) find themselves in the same node of the tree. If statehood somehow emerges without intensive agriculture, the rate matrix (Figure 1) indicates that this is evolutionarily unstable state and there is the tendency to "run away" to more stable state, that is, a combination of statehood and intensive agriculture (q24 in Figure 1). The authors interpret this high transition rate as evidence for formation of state preceding the intensification of agriculture, but it is, in fact, the other way around. This is why the results are (seemingly) at odds with other studies, including those cited in the introduction (or the aforementioned Martin et al. paper).

Similarly, that "non-grain agriculture was much more likely to be lost in states than in non-states" (line 166) does not actually suggests that "states encouraged the growing of grain at the expense of other forms of agriculture" (line 167). Pagel's test shows that the combination of statehood and non-grain agriculture is unstable, but says nothing about the direction of causality. Rather, agriculture (that is an intensive, grain agriculture), comes first, state comes later and then, grain agriculture, never loses its importance (or primacy). There is, most likely, no need to encourage or discourage production of roots or tubers. Is there some evidence to suggest otherwise? Because Pagel's test really can't decide this. That is not to say that authors interpret every single transition rate matrix incorrectly, but they have a tendency to interpret the results of Pagel's test causally, which makes the paper severely flawed.

We believe the reviewer has misinterpreted the results of the Bayesian Discrete method used in our analysis. There are obviously limits to the strength of causal claims that can be made based on any retrospective analysis. However, the restricted interpretation of our findings advocated by the reviewer is not correct, and the reviewer's claims about the likely causal direction are not supported by our findings. We have used the interpretation of the results of this method as set out by Pagel and Meade (Pagel and Meade 2006 p.820):

“The difference in the average rates of the coefficients within the rate pairs (q_{12}, q_{34}), (q_{13}, q_{24}), and (q_{21}, q_{43}) shows why models of correlated evolution predominate in the posterior sample. Rate parameters q_{13} and q_{43} spend nearly all of their time in the zero bin: ... By comparison, neither q_{24} nor q_{21} ... is found in the zero bin more than 0.34% of the time. The interpretation of these coefficients is that estrus advertisement is readily gained in the presence of a multimale mating system ($q_{24} > 0$) but has a rate of gain indistinguishable from zero in monogamous or unimale mating systems ($q_{13} \approx 0$).”

In a similar fashion, we use transition rates to compare the probability of change in one variable depending on the state of the other variable. We use this approach for each of the rate matrices. For example, the rates in Figure 5 indicate that the presence of grain agriculture makes the emergence of states possible – the rate of transition to statehood in the presence of grains (q_{34}) is positive, while the rate of transition to statehood in the absence of grains (q_{12}) is zero (lines 142-145).

Pagel and Meade (Pagel and Meade 2006 p.820) also state that:

“Figure 7 combines information on the probable ancestral states with the posterior rate distributions into a flow diagram. ... The posterior distributions of the rate coefficients suggest that mating system changed first ($q_{12} > q_{13}$) and that this created a selective force favoring estrus advertisement ($q_{24} > 0$). The alternative route in which estrus advertisement evolves first is not supported ($q_{13} \approx 0$); the path leading from the ancestral state to the intermediate state of estrus advertisement in a monogamous or single male society is not even present in 97.7% of the models.”

Analogously, in our Figure 5 the path leading from the ancestral state to the intermediate state of State without grain agriculture (q_{12}) is not present in 99.0% of the models.

The discussion is short and inadequate and does little more than just reiterating the results in the first three paragraphs. The last, fourth paragraph reads like a short advocacy for using phylogenetic comparative methods in anthropology and has little to do with the actual topic of the paper. The discussion, to put it bluntly, lacks discussion. There is no confrontation with the vast literature on the subject, not even with the few studies that are cited in the introduction. The authors gloss over the fact that their results contradict those of other authors (intensification of agriculture first, social complexity later), or discuss what „intensification“, or “complexity” are and how the way they are operationalized can affect the results.

We agree we should have explored the implications of our findings more thoroughly in the discussion and have therefore completely rewritten the discussion in the light of the reviewer’s comments. We now engage with the relevant literature on this topic and explain how and why we question some of the results of previous studies on this topic.

Since the results of the Pagel’s test are conditional on the distribution of the combination of character states at the tips of the phylogeny, it may be instructive to

look at minority cases that can help explain the general pattern. What are, for example, those states that rely on non-grain agriculture? How does taxation work in the absence of writing and what level of social complexity does it allow? Is writing ever lost? ...

At the start of each result section, we now report the binary split in the trait data and in relation to the other trait being analysed. This approach highlights the minority cases, including states that do not rely on grain agriculture. Since most of the states without grain (9/14) were part of the Atlantic-Congo language family, we pruned the global phylogeny to this family and reran the analyses. We found no correlated evolution between intensive agriculture and the emergence of states, between grain agriculture and the emergence of states, and between non-grain agriculture and the emergence of states. All the states without grain in the Atlantic-Congo language family were in Tropical Africa, this suggests that environmental or other factors may have been critical here.

Last, but not least, it would be nice to acknowledge the role of environmental constraints in formation of states. The authors make it seem like it is easy to switch from non-grain to grain agriculture once social complexity increases and with it the need for crops with taxable potential, but that is not the case, as ideal taxable crops may simply not be available. The paper does not discuss that either.

We now clearly acknowledge in the discussion that environmental factors may have slowed down or completely prevented the development of states in some areas. The lack of correlated evolution between grain agriculture and state formation in the Atlantic-Congo language family suggests that other factors were at play – at least in Tropical Africa.

In summary, while I find the topic of the paper interesting, I cannot recommend it for publication in Nature Human Behaviour.

We very much hope that the above explanation and the revisions we have made to the paper, in response to the comprehensive comments from reviewers, mean that it can now be recommended for publication in Nature Human Behaviour.

- Basava, K., H. Zhang, and R. Mace
2021 A phylogenetic analysis of revolution and afterlife beliefs. *Nat Hum Behav* 5(5):604-611.
- Bouckaert, Remco, C. Bowern, and Q. D. Atkinson
2018 The origin and expansion of Pama-Nyungan languages across Australia. *Nat Ecol Evol*.
- Bouckaert, Remco, et al.
2012 Mapping the origins and expansion of the Indo-European language family. *Science* 337(6097):957-960.
- Bouckaert, Remco, et al.
2022 Global language diversification is linked to socio-ecology and threat status. *SocArXiv*.
- Currie, Thomas E., et al.
2010 Rise and fall of political complexity in island South-East Asia and the Pacific. *Nature* 467(7317):801-804.
- Currie, Thomas E., and Ruth Mace
2009 Political complexity predicts the spread of ethnolinguistic groups. *Proceedings of the National Academy of Sciences* 106(18):7339-7344.
- Dunn, Michael, et al.
2011 Evolved structure of language shows lineage-specific trends in word-order universals. *Nature* 473(7345):79-82.
- Heggarty, P., et al.
2023 Language trees with sampled ancestors support a hybrid model for the origin of Indo-European languages. *Science* 381(6656):eabg0818.
- Her, One-Soon, et al.
2024 Early humans out of Africa had only base-initial numerals. *Humanities and Social Sciences Communications* 11(1).
- Kirby, K. R., et al.
2016 D-PLACE: A Global Database of Cultural, Linguistic and Environmental Diversity. *PLoS One* 11(7):e0158391.
- Mayshar, Joram, Omer Moav, and Luigi Pascali
2022 The Origin of the State: Land Productivity or Appropriability? *Journal of Political Economy* 130(4):1091-1144.
- Michalopoulos, S., and E. Papaioannou
2013 Pre-colonial Ethnic Institutions and Contemporary African Development. *Econometrica* 81(1):113-152.
- Pagel, Mark D., and Andrew Meade
2006 Bayesian analysis of correlated evolution of discrete characters by reversible-jump Markov chain Monte Carlo. *American Naturalist* 167(6):808-825.
- Passmore, S., et al.
2024 Global musical diversity is largely independent of linguistic and genetic histories. *Nat Commun* 15(1):3964.
- Šaffa, Gabriel, Jan Zrzavý, and Pavel Duda
2022 Global phylogenetic analysis reveals multiple origins and correlates of genital mutilation/cutting. *Nature Human Behaviour*.
- Sheehan, O., et al.
2022 Coevolution of religious and political authority in Austronesian societies. *Nat Hum Behav*.
- Skirgård, Hedvig, et al.

2023 Grambank reveals the importance of genealogical constraints on linguistic diversity and highlights the impact of language loss. *Science Advances* 9(16):eadg6175.

Stasavage, David

2021 Biogeography, writing, and the origins of the state. *In* *The Handbook of Historical Economics*. A. Bisin and G. Federico, eds. Pp. 881-902: Academic Press.

Watts, Joseph, et al.

2016 Ritual human sacrifice promoted and sustained the evolution of stratified societies. *Nature* 532(7598):228-231.

Response to the editors and reviewers (2)

Editorially we ask that you prioritise the following points:

1. It is crucial that the current work can stand alone, and for this additional methodological information is required. Please provide this, and reference a preprint of the article currently in review at Nature, and also studies that have used the tree within the main text. If the under-review paper at Nature has in the meantime been accepted for publication, please update the text to reflect this.

The reference has been updated to make clear it is a preprint, and in the Methods section (424-427) the 3 peer-reviewed studies have been referenced, and a link to the posterior distribution and all code used to generate it, inserted.

2. It is also crucial that the text (including the methods) is accessible to a broad multidisciplinary audience. Please improve the main text accordingly, in order to maximize the use-value and impact of your work for our multi-disciplinary readership.

Explanations of the methods used have been added as well as more clarity on the way the results can be interpreted. We hope that makes it more accessible to a broad audience.

Formatting of figures has been adjusted to fit with requirements.

REVIEWER COMMENTS:

Reviewer #3

Remarks to the Author:

I was very critical of this manuscript in my initial review, and I commend the authors for their thoughtful and rigorous response. The manuscript has been greatly improved.

We would like to thank the reviewer for their thoughtful and thorough review of our initial draft. We are glad that the changes in our first revision improved the paper, and hope that the additional changes in the second revision improve it further.

The main changes in this version include a more comprehensive explanation of the methods and results, particularly the selection and coding of the key variable (social

complexity) and the statistical methods used, along with clearer interpretations. New figures have been added, and the discussion has been completely rewritten. Several smaller changes further enhance the manuscript and make the presentation of the results more compelling.

I understand that a detailed description of the phylogenetic inference methods is available in an unpublished paper. The inclusion of the phylogenetic and geographic distribution of the studied variables is a significant improvement. The coevolution analysis is now described in a way that is more accessible to uninitiated readers, and the paper no longer relies so heavily on describing transition rate matrices, which made the previous version somewhat difficult to read.

The separate analysis for Niger-Congo languages in sub-Saharan Africa is a welcome addition. While it may be evident that grain is not a driver of state formation in this region (given the relative scarcity of complex societies there), including this analysis adds nuance and demonstrates attention to regional variation.

The rewritten discussion is well-crafted, and I have little to criticize. It might benefit from eliminating some repetitive statements (e.g., references to accounting for phylogenetic non-independence). A large dataset and a new supertree aside, this is a simple, elegant study and a discussion can be short.

Repetitions have been reduced.

I have a few minor comments and recommendations:

- Lines 51–54: This text seems misplaced. It addresses a methodological issue that would be better suited to the end of the introduction rather than its beginning. Most of the paragraph (lines 45–54) might fit better on page 2 when discussing the conceptualization of social complexity.

We have moved this section to the end of the introduction.

- Line 267: Citing R Development Core Team (2013) seems unnecessary in this context. If R was specifically used to create the Niger-Congo subtree, the sentence should be rephrased to clarify this. Alternatively, it may be better to remove the citation altogether.

The citation has been removed.

- I find the term "cultural phylogeny" used for the supertree somewhat misleading. The tree is based on lexical data and a set of topological constraints based on genetics and the archaeological record, correct? I assume the authors want to emphasize that this supertree is well-suited for capturing cultural macroevolution (of social complexity, music, numerals, etc.), but wouldn't "language phylogeny" be more appropriate? After all, "cultural phylogenies" can also be constructed from manuscripts, oral traditions, artifacts, etc.

"Cultural phylogeny" has been replaced with "language phylogeny" in all cases.

- The maps in the supplementary materials are of low resolution and difficult to read.

All maps in paper and supplementary information are in high resolution now.

- Transition rate matrices based on the SCCS, which were included in the main text of the previous version, should be added to the supplementary materials for completeness.

All the transition rate matrices that are not in the main text are included in the supplementary information.

Nitpicking aside, I believe this manuscript is now a strong contribution to the field, and I can recommend it for publication in Nature Human Behaviour.

That is very good to hear. Thanks again for your helpful comments.